# Does bone mineral density improve the predictive accuracy of fracture risk assessment? A prospective cohort study in Northern Denmark

Paula Dhiman,[1] Stig Andersen,[2] Peter Vestergaard,[3] Tahir Masud,[4] Nadeem Qureshi[1]

[1]Division of Primary Care, School of Medicine, University of Nottingham, Nottingham, UK
[2]Geriatric Medicine, Department of Clinical Medicine, Aalborg Universitetshospital, Aalborg, Denmark
[3]Department of Clinical Medicine and Endocrinology, Aalborg Universitetshospital, Aalborg, Denmark
[4]Geriatric Medicine, School of Medicine, University of Nottingham, Nottingham, UK

**Correspondence to**
Dr Paula Dhiman;
paula.dhiman@ndorms.ox.ac.uk

## ABSTRACT

**Objective** To evaluate the added predictive accuracy of bone mineral density (BMD) to fracture risk assessment.

**Design** Prospective cohort study using data between 01 January 2010 and 31 December 2012.

**Setting** North Denmark Osteoporosis Clinic of referred patients presenting with at least one fracture risk factor to the referring doctor.

**Participants** Patients aged 40–90 years; had BMD T-score recorded at the hip and not taking osteoporotic preventing drugs for more than 1 year prior to baseline.

**Main outcome measures** Incident diagnoses of osteoporotic fractures (hip, spine, forearm, humerus and pelvis) were identified using the National Patient Registry of Denmark during 01 January 2012–01 January 2014. Cox regression was used to develop a fracture model based on predictors in the Fracture Risk Assessment Tool (FRAX®), with and without, binary and continuous BMD. Change in Harrell's C-Index and Reclassification tables were used to describe the added statistical value of BMD.

**Results** Adjusting for predictors included in FRAX®, patients with osteoporosis (T-score ≤−2.5) had 75% higher hazard of a fracture compared with patients with higher BMD (HR: 1.75 (95% CI 1.28 to 2.38)). Forty per cent lower hazard was found per unit increase in continuous BMD T-score (HR: 0.60 (95% CI 0.52 to 0.69)). Accuracy improved marginally, and Harrell's C-Index increased by 1.2% when adding continuous BMD (0.76 to 0.77). Reclassification tables showed continuous BMD shifted 529 patients into different risk categories; 292 of these were reclassified correctly (57%; 95% CI 55% to 64%). Adding binary BMD however no improvement: Harrell's C-Index decreased by 0.6%.

**Conclusions** Continuous BMD marginally improves fracture risk assessment. Importantly, this was only found when using continuous BMD measurement for osteoporosis. It is suggested that future focus should be on evaluation of this risk factor using routinely collected data and on the development of more clinically relevant methodology to assess the added value of a new risk factor.

## Strengths and limitations of this study

► Addresses a research question recommended by The National Institute for Health and Care Excellence to investigate the added value of bone mineral density (BMD) to fracture risk prediction.

► Investigates BMD in both the commonly used, binary and continuous format.

► Presents changes in calibration, discrimination and reclassification to describe the added value of BMD.

► Uses robustly collected data from Northern Denmark, with 3.2% missing data.

► As data are from a North Danish population, with at least one fracture risk factor, this limits generalisability of the results.

occur in the USA and Europe and account for 2.8 million disability adjusted life years.[1] Further, 1.2 million disability adjusted life years are accounted for by hip fractures, which are projected to increase to 6 million by 2050.[2]

Given this burden, and treatment options for osteoporosis, identifying patients at risk of an osteoporotic fracture is high priority among health policy-makers to reduce the risk of future fracture.[3] Risk prediction tools have been developed to aid in the identification of patients at risk. For example, the Fracture Risk Assessment Tool (FRAX) and QFracture are commonly used to assess fracture risk in patients based on predefined risk factors.

Bone mineral density (BMD), a measurement used to aid diagnosis of osteoporosis, has also been identified as a fracture risk factor.[4–7] Unlike some other fracture risk factors, treatment options (eg, bisphosphonate medication) are available that reduces the fracture risk markedly when treatment is initiated based on low BMD.

## INTRODUCTION

Osteoporosis causes over 8.9 million fractures worldwide, of which over 4.5 million

English national guidelines (The National Institute for Health and Care Excellence (NICE)) for fracture risk assessment recommend treatment of osteoporosis to prevent fractures but have not included BMD as a mandatory risk factor for fracture risk prediction tools to incorporate.[8] This is partly due to the lack of robust evidence and limited generalisability of current research, which has particularly focused on evaluating BMD in postmenopausal women when evaluating the added value of BMD to existing fracture risk factors.[5–7]

NICE also recognises this gap in the evidence and have recommended research to assess the added value of BMD as a risk factor in fracture risk assessment.[9]

The aim of this study is to assess the value of BMD measurement in addition to the standard fracture risk factors used in the FRAX risk model using a robustly collected prospective cohort.

## METHODS

This paper has been written in accordance to the Transparent reporting of a multivariable prediction model for individual prognosis or diagnosis (TRIPOD) checklist.

### Patient and public involvement

Patients and the public were not involved in the development of this research question and were not involved in the design of this study.

### Study design and data source

A prospective cohort study was conducted using patients from the Aalborg University Hospital Record for Osteoporosis Risk Assessment (AURORA) dataset; patients were followed up using the National Patient Registry of Denmark.

The AURORA dataset consists of patients attending the Osteoporosis Clinic at Aalborg University Hospital after a referral from their primary care physician. A referral was offered to patients with at least one risk factor for osteoporosis (low body mass index (BMI), previous fracture, parental hip fracture, smoking status, alcohol consumption, glucocorticoid use, rheumatoid arthritis and secondary osteoporosis) or if they were aged 80 years and above. Further detail of the data collection has been described elsewhere.[10] The Danish National Patient Registry which collects inpatient and outpatient data from all Danish hospitals was linked to the AURORA dataset through unique patient identifiers.

Ethics approval was given through the Region of North Jutland's from the Danish Data Protection Agency ('paraplyanmeldelse 2008-58-0028').

### Cohort selection

Data collection for AURORA began 01 January 2010 and was collected for 3 years (up to 31 December 2012). Patients were included if they were aged 40–90 years; had a BMD T-score at the hip and were not taking any osteoporotic preventing drugs or any bone sparing drugs for more than 1 year prior to baseline.

### Primary outcome

The primary outcome measure was an incident osteoporotic fracture during follow-up (01 January 2012 to 01 January 2014), defined as a diagnosis of a fracture at the hip, spine, forearm, humerus and pelvis. Fractures at these sites resulting from traffic, work and sports-related accidents were excluded from the study. Relevant fractures were identified in the Danish National Patient Registry, using the International Statistical Classification of Diseases, 10th Revision codes (ICD-10 codes), which was developed using recognised database methodology for each fracture.[11]

### Fracture risk factors

Fracture risk factors, used in the FRAX risk prediction model, were extracted at baseline. They were: age; gender; height (m); weight (kg); previous fracture; parental history of hip fracture, current smoking status; current alcohol consumption; glucocorticoid use (currently exposed for 3+ months); rheumatoid arthritis and secondary osteoporosis (includes type 1 diabetes, osteogenesis imperfecta in adults, untreated long-standing hyperthyroidism, hypogonadism, premature menopause (<45 years), chronic malnutrition, malabsorption and chronic liver disease).

### BMD

Dual-energy X-ray absorptiometry (DEXA) scans were performed by trained technicians using Hologic Discovery A (Bedford, Massachusetts, USA). A daily quality control programme was in place and in vivo coefficient of variation using repositioning of patients was <1%. Total hip BMD was used as region of interest. BMD was added to the fracture risk prediction model twice, first, as a continuously measured T-score value, and second, as a binary risk factor, dichotomised at/above T-score threshold for osteoporosis and below threshold, −2.5 in T-score (manufacturers' normal range using normal material from T Kelly et al[12]) based on WHO classifications.[13] Calculated T-scores were gender specific.

### Statistical analysis

A complete case analysis was performed on the data; 3.2% of data were missing. The AURORA dataset was split into two using recognised methodology[14]; where a random number was assigned to patients and based on a cut-off, two-thirds was used to derive the risk models, and the remaining third was used to validate them.

### Model derivation

Three Cox proportional hazards models were developed for the primary outcome, using a complete case analysis on the derivation dataset:

  Model 1. Standard fracture risk factors only (without BMD).

  Model 2. Standard fracture risk factors (with binary BMD).

Model 3. Standard fracture risk factors (with continuous BMD).

Graphical methods were used (log–log plots) to assess the proportional hazards assumption, and risk factors violating this assumption were added to the model as a time varying covariate.

Recognised methodology used in research studies was used to build the three risk prediction models[15 16]; the Kaplan-Meier method was used to obtain 4-year fracture risk estimates for patients. Further detail on the conversion of the Cox proportional hazards models to risk prediction models has been provided in online supplementary table 1.

## Validation of models

Four-year fracture risk was calculated from each model and the predictive performance of each risk prediction model was assessed by measures describing calibration, discrimination and reclassification. These metrics were assessed using the validation cohort.

Calibration measures how well the predicted risk agrees with observed risk in the data. It plots the mean predicted and observed risk of fracture for each decile of predicted risk. The observed risk of fracture was derived from the 4-year Kaplan-Meier estimate. Good calibration indicates the predicted risk is close to the observed risk of the outcome.

Discrimination measures how well the risk prediction model differentiates between patients who have or have not observed the event in the study. This was quantified by the area under the receiver operating characteristic curve , given by Harrell's C-Index with higher values indicating better discrimination.

Reclassification tables[17] measure movement between risk categories when adding a new risk factor. Threshold for treatment at 4 years was set at a fracture risk level of 8.5%; to be comparable to the treatment threshold of 20% at 10 years. This was presented by the total per cent of patients reclassified (incorrectly and correctly) and also the Net Reclassification Index (NRI).[18 19] The NRI gives the net calculation of the changes in the right direction and a higher NRI indicates a better reclassifying model.

All analyses were carried out using Stata V.12.[20]

## RESULTS

### Characteristics of the data

The AURORA collected data on 7912 patients; 1795 patients were excluded comprising, 440 not aged between 40 and 90 years at baseline; 156 not having a recorded T-score value for the total hip at baseline and 1199 patients were taking antiosteoporotic drug therapy for more than 1 year prior to baseline.

The study sample consisted of 6117 patients; predominantly female (79.6%) and patients with a mean age of 62.9 (SD: 10.9) years. Two-thirds of this sample (n=4093) was used for the derivation dataset and one-third (n=2094) was used for the validation dataset. Table 1 presents the

baseline characteristics of the study by derivation and validation dataset and shows little difference between the datasets.

Patients in the derivation dataset had a median follow-up time of 2.30 years [IQR:1.57, 2.99] and observed 318 (7.8%) osteoporotic fractures during follow-up. Of these, 316 fractures were eligible for the analysis (two patients had a fractures on or prior to baseline and were excluded). Patients contributed 9352.8 person-years of observation, giving a total incidence rate of 337.87 per 10 000 person-years (95% CI 302.60 to 377.25).

Fractures during follow-up were predominantly found in the forearm (27.0%) and hip (17.9%). Higher fracture incidence rates were found in patients classed as osteoporotic, based on their T-score at both the total hip (809.73 per 10 000 person-years (95% CI 641.68 to 1021.78)) and spine (L1-L4) (553.59 per 10 000 person-years (95% CI 462.55 to 662.55)) (see online supplementary table 2).

### Model development

The unadjusted analysis showed statistically significant association between BMD (continuous and binary) and osteoporotic fracture (HR=0.55; 95% CI 0.50 to 0.61, p<0.001, HR=2.79; 95% CI 2.11 to 3.67, p<0.001, respectively). Significant associations with fracture were also found with age (HR=1.03; 95% CI 1.02 to 1.04, p<0.001), previous fracture (HR=3.38; 95% CI 2.69 to 4.24, p<0.001), BMI (HR=0.97; 95% CI 0.95 to 1.00, p=0.03) and gender (HR=0.73; 95% CI 0.53 to 1.00, p=0.05). Further, a time-varying effect was found in patients with a previous fracture; hazard of a subsequent fracture was highest in the first year during follow-up and decreased per year of follow-up (p<0.001).

The adjusted analysis is presented in table 2. Model 1 showed that of the standard risk factors, age and previous fracture were significantly associated with fracture; hazard of fracture increased by 2% per year increase in age (HR=1.024; 95% CI 1.013 to 1.036) and increased almost fivefold in patients with a previous fracture at time 0 years (HR=4.881; 95% CI 3.336 to 7.078).

Hazard of fracture increased by 75% for patients classed as osteoporotic by their BMD score (Model 2, HR=1.745; 95% CI 1.279 to 2.381). Hazard of fracture also decreased by 40% per SD improvement in BMD T-score (Model 3, HR=0.600; 95% CI 0.524 to 0.686).

### Model validation

The 4-year predicted risk of fracture was calculated for all patients in the validation dataset; this was compared with the observed fracture outcome within the 4-year follow-up.

#### Calibration and discrimination

Calibration plots suggested some improvement when adding BMD measurement; particularly when including continuous BMD T-score measurement (Model 3; see online supplementary figure 1).

**Table 1** Baseline characteristics of the derivation and validation datasets, including missing data

| Characteristic | Derivation (n=4093) No. | % | Validation (n=2024) No. | % |
|---|---|---|---|---|
| **Gender** | | | | |
| Female | 3266 | 79.8 | 1602 | 79.2 |
| Male | 827 | 20.2 | 422 | 20.8 |
| **Osteoporotic status (based on UK guidelines)** | | | | |
| Normal | 1886 | 46.1 | 927 | 45.8 |
| Osteopaenic | 1797 | 43.9 | 893 | 44.1 |
| Osteoporotic | 410 | 10.0 | 204 | 10.1 |
| **Previous fracture** | | | | |
| No | 2935 | 71.7 | 1423 | 70.3 |
| Yes | 1158 | 28.3 | 601 | 29.7 |
| **No. of previous fractures** | | | | |
| None | 2935 | 71.7 | 1423 | 70.3 |
| 1 fracture | 862 | 21.1 | 467 | 23.1 |
| 2–4 fractures | 270 | 6.6 | 122 | 6.0 |
| 5+ fractures | 26 | 0.6 | 12 | 0.6 |
| **Parental history of hip fracture** | | | | |
| No | 2755 | 67.3 | 1359 | 67.1 |
| Yes | 1338 | 32.7 | 665 | 32.9 |
| **Current smoking status** | | | | |
| Other (non/ex) | 3182 | 77.7 | 1529 | 75.5 |
| Smoker | 911 | 22.3 | 495 | 24.5 |
| **Alcohol consumption** | | | | |
| ≤3 units per day | 3875 | 94.7 | 1923 | 95.0 |
| >3 units per day | 218 | 5.3 | 101 | 5.0 |
| **Glucocorticoid use** | | | | |
| No | 3577 | 87.4 | 1741 | 86.0 |
| Yes | 516 | 12.6 | 283 | 14.0 |
| **Rheumatoid arthritis** | | | | |
| No | 3686 | 90.1 | 1801 | 88.0 |
| Yes | 407 | 9.9 | 223 | 11.0 |
| **Other bone affecting disease** | | | | |
| No | 2382 | 58.2 | 1139 | 56.3 |
| Yes | 1711 | 41.8 | 885 | 43.7 |
| **Secondary osteoporosis** | | | | |
| No | 3438 | 84.0 | 1689 | 83.4 |
| Yes | 655 | 16.0 | 335 | 16.6 |
| **By disease** | | | | |
| **Type 1 diabetes** | | | | |
| No | 4010 | 98.0 | 1981 | 97.9 |
| Yes | 83 | 2.0 | 43 | 2.1 |

Continued

**Table 1** Continued

| Characteristic | Derivation (n=4093) No. | % | Validation (n=2024) No. | % |
|---|---|---|---|---|
| **Osteogenesis** | | | | |
| No | 4093 | 100 | 2024 | 100 |
| Yes | 0 | 0 | 0 | 0 |
| **Hyperthyroidism** | | | | |
| No | 4089 | 99.9 | 2023 | 99.9 |
| Yes | 4 | 0.1 | 1 | 0.1 |
| **Malnutrition** | | | | |
| No | 4090 | 99.9 | 2023 | 99.9 |
| Yes | 3 | 0.1 | 1 | 0.1 |
| **Chronic liver disease** | | | | |
| No | 4006 | 97.9 | 1979 | 97.8 |
| Yes | 87 | 2.1 | 45 | 2.2 |
| **Menopause (women only)†** | | | | |
| No | 853 | 26.1 | 405 | 25.3 |
| Yes | 2413 | 73.9 | 1197 | 74.7 |
| **Premature menopause (<45 years)‡** | | | | |
| No | 1904 | 78.9 | 941 | 78.6 |
| Yes | 509 | 21.1 | 256 | 21.4 |
| | **Mean** | **SD** | **Mean** | **SD** |
| Age (years) | 62.9 | 10.9 | 63.0 | 11.0 |
| Weight (kg) | 72.1 | 15.5 | 72.2 | 15.9 |
| Missing | 47 | 1.2 | 12 | 0.6 |
| Height (m) | 1.7 | 0.1 | 1.7 | 0.1 |
| Missing | 131 | 3.2 | 61 | 3.0 |
| BMI | 26.4 | 5.0 | 26.4 | 5.1 |
| Missing | 135 | 3.3 | 63 | 3.1 |
| Hip DEXA T-score | −1.1 | 1.1 | −1.2 | 1.1 |

†Proportion out of respective number of women.
‡Proportion out of respective number of women with menopause.
BMI, body mass index; DEXA, dual-energy X-ray absorptiometry.

The largest change in discrimination was found when adding continuous BMD measurement to standard risk factors; Harrell's C-Index increased by 1.17% (table 3). However, binary BMD measurement, as a measure for osteoporotic patients, was found to reduce Harrell's C-Index by −0.65%.

### Reclassification

Reclassification tables indicated that adding continuous BMD measurement may improve classification of patients into their correct risk categories. This was not found when adding binary BMD. Table 4 presents the reclassification table for Model 1 (standard fracture risk factors

**Table 2** Multivariable analysis for osteoporotic fracture in the derivation cohort. Data are adjusted HRs and 95% CIs

| Risk factor | | Adjusted HR (95% CI) | | |
| --- | --- | --- | --- | --- |
| | | Model 1: standard risk factors only | Model 2: standard risk factors+BMD (categorical) | Model 3: standard risk factors+BMD (continuous) |
| Age (years) | | 1.024 (1.013 to 1.036) | 1.019 (1.007 to 1.031) | 1.007 (0.995 to 1.019) |
| Gender | Female | Ref | Ref | Ref |
| | Male | 0.754 (0.544 to 1.044) | 0.796 (0.573 to 1.104) | 0.851 (0.613 to 1.181) |
| BMI | | 0.978 (0.954 to 1.002) | 0.989 (0.965 to 1.013) | 1.027 (1.000 to 1.055) |
| Previous fracture | No | Ref | Ref | Ref |
| | Yes (time=0 years) | 4.881 (3.336 to 7.078) | 4.667 (3.214 to 6.778) | 4.018 (2.763 to 5.842) |
| Parental history of hip fracture | No | Ref | Ref | Ref |
| | Yes | 1.079 (0.834 to 1.397) | 1.096 (0.847 to 1.419) | 1.105 (0.854 to 1.430) |
| Current smoker | No | Ref | Ref | Ref |
| | Yes | 1.121 (0.852 to 1.475) | 1.076 (0.817 to 1.417) | 1.019 (0.774 to 1.342) |
| Alcohol consumption (>3 units/day) | No | Ref | Ref | Ref |
| | Yes | 1.414 (0.904 to 2.212) | 1.440 (0.921 to 2.252) | 1.459 (0.932 to 2.283) |
| Glucocorticoid use | No | Ref | Ref | Ref |
| | Yes | 1.080 (0.753 to 1.550) | 1.052 (0.733 to 1.510) | 1.038 (0.724 to 1.489) |
| Rheumatoid arthritis | No | Ref | Ref | Ref |
| | Yes | 1.098 (0.731 to 1.650) | 1.089 (0.725 to 1.637) | 1.116 (0.742 to 1.678) |
| Secondary osteoporosis | No | Ref | Ref | Ref |
| | Yes | 0.993 (0.729 to 1.354) | 0.966 (0.708 to 1.317) | 0.911 (0.667 to 1.243) |
| Osteoporotic | No | Ref | Ref | Ref |
| | Yes | – | 1.745 (1.279 to 2.381) | – |
| Hip DEXA T-score (SD) | | – | – | 0.600 (0.524 to 0.686) |
| Previous fracture (TVC*) | | 0.635 (0.489 to 0.826) | 0.639 (0.492 to 0.830) | 0.644 (0.495 to 0.837) |

Data are adjusted HRS and 95% CIs.
*TVC, value is interaction effect and 95% CI.
BMD, bone mineral density; BMI, body mass index; DEXA, dual-energy X-ray absorptiometry; TVC, time-varying covariate.

only) and Model 3 (standard risk factors with continuous BMD), using the 8.5% prespecified risk threshold.

Of the 1960 patients in the validation dataset, 27% (n=529) were reclassified into a different risk category when including continuous BMD into fracture risk prediction. Two per cent (9/529) were found to be reclassified correctly into a higher risk group and 55% (292/529) were reclassified correctly into a lower risk group; indicating 22% (292/1342) of patients at high risk in Model 1, not accounting for BMD measurement, were no longer at high risk. The net reclassification improvement when

adding continuous BMD to standard risk factors, was 0.03, which resulted from increased specificity (non-event NRI=4%) and decreased sensitivity (event NRI=−1%) from Model 1 (table 5).

## DISCUSSION
### Summary of findings
BMD showed significant association with fracture risk with a 40% decrease for each SD rise in BMD. However, this resulted in small improvements in calibration,

**Table 3** Harrell's C-Index for models 1, 2, and 3

| Model | Harrell's C-Index | Change in Harrell's C-Index (% change)* |
| --- | --- | --- |
| Model 1: Standard fracture risk factors only (without BMD) | 0.764 (0.718 to 0.810) | – |
| Model 2: Standard fracture risk factors only (with binary BMD) | 0.759 (0.712 to 0.806) | −0.005 (-0.65%) |
| Model 3: Standard fracture risk factors only (with continuous BMD) | 0.773 (0.732 to 0.814) | 0.009 (1.17%) |

*All change is measures against Model 1.
BMD, bone mineral density.

**Table 4** Risk reclassification table comparing Model 1 (standard fracture risk factors alone) to Model 3 (standard fracture risk factors with continuous BMD measurement), using a clinical 8.5% risk cut-off

| | | | Model 3: SRF with continuous BMD | | Total | Total no.(%) reclassified |
| | | | <8.5% | ≥8.5% | | |
|---|---|---|---|---|---|---|
| Model 1: SRF without BMD | <8.5% | No. | 391 | 227 | | 227 (36.7) |
| | | % | 63.3 | 36.7 | | |
| | | No. of events | 5 | 9 | 618 | |
| | | No. of non-events | 386 | 218 | | |
| | | Observed event rate | 1.3% | 4.0% | | |
| | ≥8.5% | No. | 302 | 1040 | | 302 (22.5) |
| | | % | 22.5 | 77.5 | | |
| | | No. of events | 10 | 121 | 1342 | |
| | | No. of non-events | 292 | 919 | | |
| | | Observed event rate | 3.3 | 11.6 | | |
| | Total | | 693 | 1267 | 1960 | 529 (27.0) |

BMD, bone mineral density; SRF, standard risk factors.

discrimination and reclassification. The C-index estimate was slightly higher with continuous BMD, but this increase is not conclusive given the width of the CIs. Despite the limited improvement found of 1% in discrimination when adding continuous BMD, reclassification tables showed 57% of reclassified patients moving into their correct risk group through improved specificity. Importantly, no improvement was found when adding BMD in a binary format.

Our findings are consistent with and corroborate with current literature.[7 21 22] Specifically, a study conducted in the Netherlands with 4-year follow-up, investigating the added value of BMD for hip fractures risk, found modest improvement in predictability.[21] Further, two more recent studies also indicated limited added value of BMD to fracture risk prediction.[7 22]

### Strengths and limitations
#### Answering evidence gap
To our knowledge, this is the first study to investigate the added value of BMD in a binary and continuous format to standard fracture risk factors. Further, it is based on a larger sample size than other studies investigating BMD in addition to FRAX®.[7 21 22] It helps inform the NICE research recommendation to assess the added value

of BMD to routine fracture risk assessment in primary care.[23] It further highlights that the more commonly used for treatment decision-making, binary format of BMD resulted in a loss of predictability in fracture risk prediction based on comparable measures for discrimination and reclassification.

#### Robustness of data
The prospective cohort was well populated with key standard risk factors recorded: BMI, smoking status and alcohol consumption and personal and parental fracture history. Other than 3.2% of missing data for BMI, in 6117 patients, complete data were collected for all risk factors (including BMD T-score recorded at the total hip). Further, the cohort was linked to a national robust electronic health records. This Danish National Patient Registry allowed for outcome fracture to be identified and also provided data on the mechanism for the fracture; this helped more accurately phenotype osteoporotic fractures.

#### Generalisability
The generalisability is affected in a few ways. First, the findings are based on a Danish cohort. Second, AURORA data were collected from patients who presented to their doctor with at least one fracture risk factor and were referred to the osteoporosis clinic; this led to a biased study sample with a higher risk of a fracture and increased age. This could overestimate fracture risk among patients in a primary care setting.

#### Methodology
Due to the increased age of the sample, death becomes a competing risk. However, information on death was not collected and could not be retrieved. This limited the analysis of the data as competing risks could not be accounted for which may again lead to an overestimation

**Table 5** Summary of NRI for all comparisons between developed fracture risk prediction models

| Comparison | Event NRI | Non-event NRI | Overall NRI |
|---|---|---|---|
| Model 1 versus model 2 | −3.45% | 2.09% | −0.01 |
| Model 1 versus model 3 | −0.69% | 4.08% | 0.03 |

NRI, Net Reclassification Index.

of fracture risk.[24] However, as an independent study primarily assessing the added value of BMD through deriving and validating the fracture risk prediction models, this bias would be present in both analyses to compare derived risk models with and without BMD measurement.

The FRAX® risk algorithm has not yet been published, therefore FRAX® estimates could not be directly calculated for the cohort. Instead, the FRAX® risk model was recalibrated on the dataset with and without BMD added. Further, fracture outcomes in this study included pelvic fractures which are increasingly recognised as low-trauma fragility fractures [25] and used BMD taken at the total hip instead of at the femur neck as it is the gold standard in Denmark.[26]

Internal validation was performed to validate the derived risk prediction models. This may lead to overoptimistic results of the performance of the risk models.[14] To account for this limitation, a commonly practised method which randomly assigns patients to the derivation and validation datasets was used; further, a similar 1:2 ratio was also used to split the data.[27–29]

The study had a 4-year follow-up which is shorter than other recognised risk models. To account for this, we adapted the 20% clinical risk threshold for 10-year fracture estimates to 8.5% for 4-year fracture estimates, assuming that risk is constant over time.[30 31]

Traditional methodology assessing the added value to risk factors to existing risk prediction models are criticised to be insensitive to change, to lack interpretability.[32–35] This was shown when finding a 1% change in Harrell's C-Index and overlapping CIs between models, limiting the interpretability of results. Reclassification analysis was thus also used to provide more clinically interpretable results.

## Clinical implications

The most notable clinical implication is the more routine use of BMD measurement for fracture risk assessment. Further, evidence suggests continuous BMD adds better predictability compared with the binary format.

## Future research

Further research is recommended to evaluate the added value of BMD to fracture risk prediction, in particular, in addition to QFracture risk factors and using primary care routinely collected data. However, a brief interrogation into the Clinical Practice Research Datalink, a routinely collected UK primary care database, showed poor availability of BMD measurement in patient records, and thus, strong limitations to potential analyses. Less than 1% of patients had BMD recorded from a sample of 60 658 patients aged 40–90; not on any osteoporotic treatment and with complete data for age, gender, BMI, smoking status and alcohol consumption. Thus, prior to UK analysis, BMD recording in primary care databases needs to improve.

Methodologically, as well as assessing the added value of BMD to standard risk factors, we should also explore the option to replace existing fracture risk factors with the BMD measurement; this has rarely been explored in the literature but should be considered in future analyses. We also recommend research to investigate the added value of BMD in a potentially more natural, three-group format of BMD (osteopaenic, normal, osteoporotic).

In addition, further research is recommended to develop current methodology used to assess the added value of BMD to provide more clinically relevant results, such as cost implications and to allow for better comparability between new risk factors with respect to their added value, thus improving decision-making.

## CONCLUSION

Continuous BMD marginally improves fracture risk assessment. Importantly, this was only found when using continuous BMD measurement for osteoporosis. It seems that prediction models for fragility fracture risk may be improved only marginally, using present risk factor assessment and evaluations. It is suggested that future focus should be on additional risk factors and on the development of more clinically relevant methodology to assess the added value of a new risk factor.

### Transparency declaration

The lead author (PD) affirms that this manuscript is an honest, accurate and transparent account of the study being reported; that no important aspects of the study have been omitted and that any discrepancies from the study as planned (and, if relevant, registered) have been explained.

**Contributors** PD wrote the statistical analysis plan, cleaned and analysed the data, and drafted and revised the paper. PV and SA provided the AURORA dataset for analysis and linked patients to the National Patient Registry of Denmark, they also reviewed and revised the draft paper. NQ and TM provided clinical expertise, and reviewed and revised the draft paper.

**Funding** This paper presents independent research and was funded by the National Institute for Health Research (NIHR), School for Primary Care Research (SPCR).

**Disclaimer** The views expressed are those of the author(s) and not necessarily those of the NIHR, the NHS or the Department of Health.

**Competing interests** No, there are no competing interests for any author.

**Patient consent** Detail has been removed from this case description/these case descriptions to ensure anonymity. The editors and reviewers have seen the detailed information available and are satisfied that the information backs up the case the authors are making.

**Ethics approval** Ethics approval was given through the Region of North Jutland's from the Danish Data Protection Agency ('paraplyanmeldelse 2008-58-0028').

**Provenance and peer review** Not commissioned; externally peer reviewed.

**Data sharing statement** Technical appendix and statistical code is available from the corresponding author at paula.dhiman@ndorms.ox.ac.uk. Due to restrictions by the Danish Data Protection Agency, data can only be shared on an aggregated level and by special permission.

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
