## [Reviewer comments · BMJ Open]

ARTICLE DETAILS

TITLE (PROVISIONAL)	Does Bone Mineral Density Improve the Predictive Accuracy of Fracture Risk Assessment? A Prospective Cohort Study in Northern Denmark.
AUTHORS	Dhiman, Paula; Andersen, Stig; Vestergaard, Peter; Masud, Tahir; Qureshi, Nadeem

VERSION 1 – REVIEW

REVIEWER	Jose Antonio P da Silva Faculty of Medicine, University of Coimbra, Portugal
REVIEW RETURNED	31-Aug-2017

GENERAL COMMENTS	This paper addresses the contribution of BMD to fracture risk prediction. Although methodology is generally clear, sound and appropriate, and some limitations are discussed, there are some drawbacks that need clarification. Major issues: 1. Although the authors refer to FRAX and limit their investigation of clinical risk factors to those included in this algorithm, it is not clear why1.A. The risk estimates of FRAX itself were not used1.B. The definition of major fractures differs, although slightly, from the one used in FRAX (No pelvis fractures and only Clinical vertebral fractures)1.C. Why total hip BMD was used instead of Neck of femur This needs discussion as FRAX has been validated in so many countries including Denmark and already predicts the use of BMD. This would certainly increase the generalizability of the results, a correct cause of concern to the authors.2. The mean follow-up of participants does not result clear from reading methods. In fact some data suggest that it was, on average, of only 1,53 years (9353person/years of observation in 6117 patients - Page 8)3. The choice of the 8.5 risk threshold to define risk categories is based on the assumption that the yearly risk is similar between 0 and 10 years of follow-up - Please comment4. The novelty of your observations should be addressed and discussed: many publications, including the meta-analyses supporting FRAX indicate that BMD adds to the risk estimation. Several analyses of the performance of FRAX with and without BMD suggest the same, even if at a small difference.5. Considering BMD resulted, in the best scenario, in an increase of Harrel's C of 1% - Please discuss the clinical relevance of this difference
---

	6. Dychotomizing risk at exactly 8.5% may give an artificial perspective of the impact of considering BMD, as many participants may move from one to the other side of the cut-off without major changes in risk. Please comment on this, Please consider adding an in-between risk class of say 7.5 to 9.5% (or another interval considered clinically significant 5. The idea a of substituting the consideration of (free and immediately accessible) clinical risk factors by BMD needs reconsideration, or at least serious discussion, in face of the points raised above. Minus points 7. The reference made in the abstract to FRAX (page 3, Ln18-19) needs to be revised not to induce the reader to belief that FRAX estimates were used 8. Medication used after baseline needs to be described in summary and its influence explored 9. please specify if gender specific T scores were used, and discuss the rationale of the choice, faced with the choices made by FRAX 10. In Page 10 Ln49 you hyPOthyroidism isn listed as risk factor for OP. Do you mean HyPER? end
--	---

REVIEWER	Eugene McCloskey University of Sheffield, UK
REVIEW RETURNED	01-Sep-2017

GENERAL COMMENTS	This is a prospective study of fracture prediction within the setting of a secondary care referral centre in North Denmark. It concludes that BMD, particularly when treated as a continuous variable, can enhance fracture prediction when combined with clinical risk factors. Despite the study being designed to answer a question raised by NICE in the UK, the question itself reflects a misunderstanding of the development of the FRAX tool that optionally contains BMD input. Indeed, the whole premise underlying the inclusion of clinical risk factors in FRAX was that they should be at least partly independent of BMD, demonstrated in several meta-analyses, so that the finding that BMD adds value is not at all surprising. Nonetheless, the study has merits and I would make the following comments: a) The biggest limitation by far is the sample size and thus the number of incident fractures. This will limit the ability of the analyses to detect significant prediction by some of the variables included in the FRAX tool; the conclusion that BMD can replace such risk factors is therefore incorrect. It is certainly not accurate to raise this suggestion without addressing the limitations. b) The finding that use of continuous BMD is better than dichotomous BMD is not surprising and again is driven by statistical/mathematical considerations, whereby a categorical variable either has to have a high prevalence or a very high gradient of risk (or ideally both) before it can impact much on AUROC. The authors quite rightly combine this comparison with reclassification analysis. c) If the purpose of a test, such as BMD, is to aid in the classification of subjects, then its utility is maximal when targeted to those at or near the intervention or diagnostic threshold, rather than in all subjects. I would welcome an additional analysis to see what thresholds of risk (without BMD) would have captured for example 90% of those reclassified, so that one could make better use of the
---

	resource. d) Finally, the real question that arises is how does BMD perform when combined with risk factors used in other risk assessment tools where BMD has not been included, e.g. Qfracture, and where their independence from BMD has not been demonstrated previously. The paper would be much strengthened by addressing this question.
--	---

REVIEWER	Kristin Sainani Stanford University, USA
REVIEW RETURNED	29-Sep-2017

GENERAL COMMENTS	This study aimed to evaluate the added predictive accuracy of adding hip BMD T-scores to an established fracture risk assessment tool. The manuscript has many strengths: The study is prospective; the paper is concise and well-written; the authors have included discrimination, calibration, and reclassification metrics; and the question is of high clinical importance. The study shows that there is some benefit to adding BMD as a continuous variable to current fracture risk models. However, the added predictive accuracy is relatively small and has to be weighed against the added costs of BMD testing. Specific comments:  1. The authors are to be commended for including both a derivation and validation cohort. They have appropriately used the validation cohort when calculating reclassification statistics. However, it appears that they used the derivation cohort when calculating discrimination and calibration statistics. This needs to be fixed—all performance metrics should be calculated using the validation cohort only. 2. The authors need to be more careful in interpreting and reporting the results for “previous fracture” because of the inclusion of a time*previous fracture term in the model. For example, for model 1, in the tables and text, they imply that the hazard ratio for previous fracture is 4.88. This is incorrect. The hazard ratio is only 4.88 when time=0. In fact, the hazard ratio for previous fracture changes with time: $HR = \exp(\ln(4.88) - .446 * \text{time})$. So, the hazard ratio is 3.12 when time=1 year; 2.0 when time=2 years; 1.28 when time=3 years; and 0.82 when time=4 years. Also, exponentiating the beta for the time*previous fracture coefficient does not give a HR. The interaction term must be combined with main effects to make meaningful hazard ratios. So, for example, the reported HR of 0.64 for model 1 (in Table 2) is meaningless. The best way to report time-varying covariates is to report the HR for several example time points (e.g., at baseline, 1 year, 2 years). 3. Only a small number of people are osteoporotic (10% of the sample). So, it’s not surprising that the binary BMD variable (osteoporotic yes/no) does not improve predictive accuracy. What happens if you include a 3-level categorical variable for BMD: normal, osteopenic, osteoporotic? This seems like the more natural way to treat BMD as a categorical variable than a simple osteoporotic/not osteoporotic split. 4. Under “Model development,” this needs to be rephrased: “Adding binary and continuous BMD to standard risk factors led to a 75% increased hazard of fracture, whilst adding continuous BMD T-score led to a 40% lower hazard per SD improvement...” Though I understand the intent, in fact, “adding variables to the model” does not increase a person’s hazard of fracture. 5. Table 1: need to give units for alcohol consumption as <3 and >3 units PER DAY rather than just <3 and >3 units.
---

	6. Please reduce the number of decimal places in Tables 1, 3 and 4, e.g., .759 rather than .7592 and 36.7% rather than 36.73%. This makes the tables easier to read. 7. What were the standard errors and p-values for the NRIs? 8. The NRI for events can be interpreted as the increase in sensitivity and the NRI for non-events can be interpreted as the increase in specificity. Please note this for readers to help draw attention to the fact that the inclusion of BMD is helping to improve specificity but not sensitivity. 9. The authors have done a nice job discussing the issue of generalizability in their discussion. I'd like to see a little more discussion about the tradeoffs between slightly improved accuracy and cost. The standard model does pretty well with variables that are inexpensive to measure. This is largely because BMI is tightly linked to BMD, so including BMI in the model gives indirect information about BMD. Is the added cost of BMD screening offset by the improved specificity of the new model? Some further discussion of this is warranted.
--	---

REVIEWER	John Kornak University of San Francisco, California. USA
REVIEW RETURNED	01-Oct-2017

GENERAL COMMENTS	Statistics in the paper were thorough. Good checking of modeling assumptions in Survival analysis, and good practice of providing estimates, confidence intervals and p-values throughout. However, the interpretation of model fit quality differences seems to be way over-played. Minor comments P 7. line 21/22 looks like there is a typographical error around the word "randomly" P 11 line 32/33 -- HR and CI needs an extra decimal point of accuracy (currently from rounding, the HR could be between 1.015, to 1.025; a 60% range!) -- or perhaps even better for clinical relevance would be to provide HR per 10 year difference in age. P 11 line 41 -- Do not use insignificant to mean did not reach statistical significance. Not statistically significant just means there is not enough evidence to reject the null hypothesis; but absence of evidence is not the same as evidence of absence. The authors would need to argue that the estimated HR and the complete range of the confidence interval are too small to be clinically relevant in order to argue insignificance. P 13 lines 10 and 11 -- based on the results I don't think this statement has been conclusively demonstrated and in particular for BMDcat P 13 lines 13 to 16 -- authors need to note that these differences do not represent reliable improvements of one model over another. The differences between models are very small compared to the width of the confidence intervals. P 13 lines 41/42 I don't think this has been demonstrated. P 13 line 40 -- NRI of 0.03 is small -- could it even be a chance improvement? Table 4 doesn't seem to show an overwhelming improvement. Similarly Table 5 NRIs are all small P 15 line 7/8 I think this conclusion is overstated. P 16 line 12/13 -- ratio should be 1:2 rather than 1:3 based on earlier text. (2/3 derivation and 1/3 validation)
---

VERSION 1 – AUTHOR RESPONSE

Responses to Reviewers Comments

Comment	Response	Page No
Editor Comments		
You state that you adhered to the TRIPOD checklist - please include a completed copy of the TRIPOD checklist as a supplementary file.	Thank you for this comment. We did attach the completed checklist as a supplementary file. We shall re-attach with the resubmission	Attached
The Discussion section could be improved: what does this study add to the literature? How do the results compare to previous studies?	Thank you for this comment the Discussion has been extensively revised. We have included responses to the reviewers' comments below and have also added how the study adds/compares to the literature. Specifically, our findings were consistent with current literature; a study conducted in the Netherlands with 4 year follow up, investigating the added value of BMD for hip fractures risk found modest improvement in predictability.	p15-17
Reviewer 1: This paper addresses the contribution of BMD to fracture risk prediction. Although methodology is	Thank you for highlighting the strengths of this study and our methodology. We have supplied clarification to your points, raised below.	-

generally clear, sound and appropriate, and some limitations are discussed, there are some drawbacks that need clarification.		
1. Although the authors refer to FRAX and limit their investigation of clinical risk factors to those included in this algorithm, it is not clear why	Thank you for this query. We arrived at using FRAX for the investigation based on the following reasons: 1) This study was motivated by research recommendations listed in the NICE guidance (https://www.nice.org.uk/researchrecommendation/bone-mineral-density-bmd-with-frax-what-is-the-added-prognostic-value-of-bmd-in-the-assessment-of-fracture-risk-with-frax), which recommended investigation of the added value of BMD to fracture risk prediction. Based on this guidance, we wanted to investigate the added value of BMD on a risk prediction model commonly used as part of UK guidance - QFracture and FRAX. 2) FRAX optionally includes BMD as an additional risk factor and hence provides an ideal platform to assess its added value.	p17

	3) We also found it challenging to identify a dataset with all the risk factors used in QFracture, and which also had BMD recorded. We initially attempted to use data extracted from the Clinical Practice Research Datalink to assess the added value of BMD, which would have allowed us to evaluate BMD to QFracture and FRAX. We extracted a sample from 2004 to 2014, of patients aged 40-90 years who were registered at their practice for at least 1 year, and excluded patients on osteoporotic treatment/bone sparing medication, but, unfortunately, didn't have complete data recorded on core fracture risk factors (age, gender, height, weight, current smoking status, and alcohol consumption) within 12 months prior to baseline.	
--	---	--

	A cohort of 60,658 patients was extracted, of which 3 patients had BMD recorded. We have added this to the discussion ('Discussion–Future Research').	
1.A. The risk estimates of FRAX itself were not used	Thank you for this comment. We agree that ideally FRAX estimates should have been calculated for the cohort. However, the FRAX algorithm has not been published, so	p16

	FRAX estimates could not be calculated. We therefore, recalibrated the FRAX algorithm on the AURORA dataset, with and without the BMD. We have included this explanation in the discussion ('Discussion-Methodology').	
1.B. The definition of major fractures differs, although slightly, from the one used in FRAX (No pelvis fractures and only Clinical vertebral fractures)	Thank you for raising this issue. We accept that pelvis fractures are not currently used in the FRAX risk model, however, current evidence increasingly recognises pelvis fractures as low trauma fragility fractures (1). Given this information was available and that FRAX risk prediction model was recalibrated on this dataset, we utilised this data in the presented risk prediction model. We have added this discrepancy to the discussion ('Discussion – Methodology').	p16
1.C. Why total hip BMD was used instead of Neck of femur This needs discussion as FRAX has been validated in so many countries including Denmark and already predicts the use of BMD. This would certainly increase the	Thank you for this question, we agree that the FRAX model uses BMD from the neck of the femur however, this data was not available in the AURORA dataset. We had data on spine and total hip BMD only. This is because total hip is considered the gold standard in Denmark (2) and it was decided to use this in the evaluation of patients at the	p16

generalizability of the results, a correct cause of concern to the authors.	Osteoporosis Clinic in North Denmark. We have added this to our discussion ('Discussion – Methodology').	
2. The mean follow-up of participants does not result clear from reading methods. In fact some data suggest that it was, on average, of only 1,53 years (9353person/years of observation in 6117 patients - Page 8)	Thank you for this comment, however, we are a little unsure as to what the reviewer means here. We hope the response below satisfies the comment. The 9353 person years was calculated for the derivation dataset, which included 4,093 patients and 316 incident fractures. We feel it is not appropriate to present the mean follow up here as it violates the assumptions of survival analysis. Instead the median follow up time would be more appropriate and was calculated to be 2.30 years [1.57, 2.99], which includes patients who did and did not experience a fracture. Given the overlap in the recruitment and follow up periods, this resulted in less than 4 years follow up for some patients. These were treated as censored patients in the analysis and had a median follow up time of 2.37 years [1.71, 3.05].	p9

	We have added this information in the results ('Results -	
	Characteristics of the data').	
3. The choice of the 8.5 risk threshold to	Thank you for raising this point. Indeed, the underlying	p16
define risk categories is based on the	assumption of calculating the 4 year risk threshold of 8.5%	
assumption that the yearly risk is similar	from the 10 year risk threshold of 20%, is that the yearly	
between 0 and 10 years of follow-up -	risk is similar.	
Please comment		
	We have added this to the discussion ('Discussion -	
	Methodology').	
4. The novelty of your observations	Thank you for your comment. We agree, with the	-
should be addressed and discussed:	reviewer. However, as mentioned above this study was	
many publications, including the meta-	motivated by NICE. The NICE guideline development	
analyses supporting FRAX indicate that	group, following comprehensive review of the literature,	
BMD adds to the risk estimation.	listed research recommendations to investigate the added	
Several analyses of the performance of	value of BMD to fracture risk prediction. This suggests	
FRAX with and without BMD suggest	there is still a recognised gap in the evidence around this,	
the same, even if at a small difference.	and that further work is required.	
	In relation to the 'novelty' of the observations, this	
	addresses the poorer predictability when using binary	
	BMD, as opposed to the continuous format.	

5. Considering BMD resulted, in the best	Thank for this comment. We accept that a 1% increase in	p15
scenario, in an increase of Harrell's C of	Harrell's C-Index is small, however this is a common	
1% - Please discuss the clinical	limitation of the measure itself for not being sensitive	
relevance of this difference	enough to change and also difficult to interpret in clinical	
	context (3, 4). Further, a 1% change has also shown to be	
	relevant in other studies such as when adding parental	
	history and c-reactive protein to the Reynolds Risk	
	Algorithm (5).	
	Further, this small improvement was accounted for by also	
	presenting the more clinically relevant reclassification	
	analysis, which showed 27% of patients were reclassified	
	on addition of continuous BMD. Of these patients, a total	
	of 301 (57%) were reclassified correctly.	
	We have added this to the discussion ('Discussion –	
	Summary of findings')	
6. Dychotomizing risk at exactly 8.5%	Thank you for this comment. As described in the paper the	
may give an artificial perspective of the	8.5% risk threshold for 4 year risk estimates was calculated	
impact of considering BMD, as many	in line from the 20% risk threshold for 10 year risk	
participants may move from one to the	estimates.	
other side of the cut-off without major		
changes in risk. Please comment on this,	We are concerned that reasons would be unclear and not	

Please consider adding an in-between risk class of say 7.5 to 9.5% (or another interval considered clinically significant)	justifiable to produce results for other risk thresholds (including 7.5% and 9.5%).	
7. The idea a of substituting the consideration of (free and immediately accessible) clinical risk factors by BMD needs reconsideration, or at least serious discussion, in face of the points raised above.	Thank you for this comment. The intention for this part of the analysis was more in line to explore methodological advancements and the impact of adding BMD beyond changes in predictability.	p3, p4, p7, p11, Table 2, Table 3, Table 5, p15-17

	We agree, however that given the limitations of the analysis and publication strategy of the BMJ Open, this is not applicable here and have removed this analysis from the paper.	
8. The reference made in the abstract to FRAX (page 3, Ln18-19) needs to be revised not to induce the reader to belief that FRAX estimates were used	Thank you for raising this point. We have modified the abstract to reflect this.	p3
9. Medication used after baseline needs to be described in summary and its influence explored	Thank you for this suggestion. Initiation of anti-osteoporotic drug based on BMD could diminish the strength of BMD measurement in the data. Unfortunately, we did not have access to these data. Therefore, we could not explore this in the analysis.	-

	We did however, try to minimise the impact of this by excluding patients on any osteoporotic or bone sparing medication at baseline.	
10. please specify if gender specific T scores were used, and discuss the rationale of the choice, faced with the choices made by FRAX	Thank you for this comment. Gender specific T-scores were used which was in accordance to FRAX. We have added this information into the methods ('Methods – Bone Mineral Density').	p7
11. In Page 10 Ln49 you hyPOthyroidism isn listed as risk factor for OP. Do you mean HyPER?	Thank you for this comment. Yes, this was meant to be hyperthyroidism. We have amended this in the paper	Table 1 - p10
Reviewer 2: This is a prospective study of fracture prediction within the setting of a secondary care referral centre in North Denmark. It concludes that BMD, particularly when treated as a continuous variable, can enhance fracture prediction when combined with clinical risk factors. Despite the	Thank you for these comments and highlighting the robust development of our FRAX based model. Incorporating clinical factors and BMD. As mentioned previously, our study aims to respond to a NICE research recommendation – ‘BMD with FRAX: What is the added prognostic value of BMD in the assessment of fracture risk with FRAX?’	-

study being designed to answer a question raised by NICE in the UK, the question itself reflects a misunderstanding of the development of the FRAX tool that optionally contains BMD input. Indeed, the whole premise underlying the inclusion of clinical risk factors in FRAX was that they should be at least partly independent of BMD, demonstrated in several meta-analyses, so that the finding that BMD adds value is not at all surprising. Nonetheless, the study has merits and I would make the following comments:	Our observation of this was that as NICE listed the evaluation of the added value of BMD in their research recommendation in light of the development of FRAX and evidence showing the added value of BMD, further research and evidence was needed for better implementation of BMD in clinical practice.	
1. The biggest limitation by far is the sample size and thus the number of incident fractures. This will limit the ability of the analyses to detect significant prediction by some of the	Thank you for this comment. As described above, the purpose for this part of the analysis was more in line to explore methodological	p3, p4, p7, p11, Table 2, Table 3,
variables included in the FRAX tool; the conclusion that BMD can replace such	advancements and the impact of adding BMD beyond changes in predictability.	Table 5, p15-17

risk factors is therefore incorrect. It is certainly not accurate to raise this suggestion without addressing the limitations.	We agree however, that sample size, a short follow up, and few incident fractures do pose limitations to the analysis and findings, especially in relation to the replacement of existing risk factors with BMD. Given this advice from the reviewers, we have removed the latter analysis.	
2. The finding that use of continuous BMD is better than dichotomous BMD is not surprising and again is driven by statistical/mathematical considerations, whereby a categorical variable either has to have a high prevalence or a very high gradient of risk (or ideally both) before it can impact much on AUROC. The authors quite rightly combine this comparison with reclassification analysis.	Thank you for this comment. Indeed, it was expected that continuous BMD measurement would perform better than dichotomised BMD. However, the interesting result was that dichotomised BMD added no improvement to the fracture risk prediction model. The reclassification also confirmed this analysis.	-
3. If the purpose of a test, such as BMD, is to aid in the classification of subjects, then its utility is maximal when	Thank you for this comment. Indeed, it is very interesting to see the impact of adding BMD and the movement of patients between their risk categories. As well as	p17

targeted to those at or near the intervention or diagnostic threshold, rather than in all subjects. I would welcome an additional analysis to see what thresholds of risk (without BMD) would have captured for example 90% of those reclassified, so that one could make better use of the resource.	affecting the predictive accuracy of the fracture risk model, which we described using current and new (reclassification) methods, it also may affect the risk threshold used for treatment. We are in the process of exploring this approach for publication in a future paper. The potential role of this extra analysis is added to future research section on the discussion ('Discussion – Future Research).	
4. Finally, the real question that arises is how does BMD perform when combined with risk factors used in other risk assessment tools where BMD has not been included, e.g. Qfracture, and where their independence from BMD has not been demonstrated previously. The paper would be much strengthened by addressing this question.	Thank you for this comment. Again, the reviewer raises a very valid point. Initially, we also planned to interrogate the QFracture risk model which includes many risk factors, but not BMD. However, we found it difficult to identify a suitable dataset which collects both QFracture risk factors and BMD measurement. We initially attempted to use data extracted from the Clinical Practice Research Datalink (a similar dataset to QResearch, in which the QFracture risk model was developed) to assess the added value of BMD. We extracted a sample from 2004 to 2014, of patients	p17

	aged 40-90 years who were registered at their general practices for at least 1 year, and excluded patients on osteoporotic treatment/bone sparing medication, and didn't have complete data recorded on core fracture risk factors (age, gender, height, weight, current smoking status, and alcohol consumption) within 12 months prior to baseline. A cohort of 60,658 patients were extracted, of which 3 patients had BMD recorded. Thus the analysis was not possible. We have included the reviewers' suggestion in the discussion ('Discussion – Future Research').	
--	--	--

Reviewer 3: This study aimed to evaluate the added predictive accuracy of adding hip BMD T-scores to an established fracture risk assessment tool. The manuscript has many strengths: The study is prospective; the paper is concise and well-written; the authors have included discrimination, calibration, and reclassification	Thank you for your kind comments on the strengths of the paper. We agree that an economic appraisal of the costs and benefits of BMD would be beneficial to the existing evidence base-, and justifies a separate publication. Based on a recently completed doctorate thesis], we are preparing a related health economics paper. The potential role of this extra analysis is added to future research section on the discussion ('Discussion – Future	p17

metrics; and the question is of high clinical importance. The study shows that there is some benefit to adding BMD as a continuous variable to current fracture risk models. However, the added predictive accuracy is relatively small and has to be weighed against the added costs of BMD testing. Specific comments:	Research).	
1. The authors are to be commended for including both a derivation and validation cohort. They have appropriately used the validation cohort when calculating reclassification statistics. However, it appears that they used the derivation cohort when calculating discrimination and calibration statistics. This needs to be fixed—all performance metrics should be calculated using the validation cohort only.	Thank you for your kind comments and recognising the methodological robustness of the analysis. We apologise for the confusion, we did follow standard analytic approaches with development of the risk models using the derivation cohort, and assessed the calibration, discrimination, and reclassification using the validation cohort only. We have added a sentence to the methods section, to clarify that this was the analytic approach ('Methods – Model Derivation and Validation of Models').	p7-8
2. The authors need to be more	Thank you for this comment and your calculations.	p11,

careful in interpreting and reporting the results for “previous fracture” because of the inclusion of a time*previous fracture term in the model. For example, for model 1, in the tables and text, they imply that the hazard ratio for previous fracture is 4.88. This is incorrect. The hazard ratio is only 4.88 when time=0. In fact, the hazard ratio for previous fracture changes with time: $HR = \exp(\ln(4.88) - .446 * \text{time})$. So, the hazard ratio is 3.12 when time=1 year; 2.0 when time=2 years; 1.28 when time=3 years; and 0.82 when time=4 years. Also, exponentiating the beta for the time*previous fracture coefficient does not give a HR. The interaction term must be combined with main effects to make meaningful hazard ratios. So, for example, the reported HR of 0.64 for	You are correct, the interaction between previous fracture and time can be presented in this way. We have clarified in the results that the HR of 4.88 refers to time = 0 years. However, as the HR is calculated for any previous fracture at any time and more distant fractures may be associated with less excess fracture risk than recent, we feel that this may not be the best way to describe it in this study. Thus we have kept in line with previous literature and kept to the current presentation in the paper.	Table 2
---	--	----------------

model 1 (in Table 2) is meaningless. The best way to report time-varying covariates is to report the HR for several		
example time points (e.g., at baseline, 1 year, 2 years).		
3. Only a small number of people are osteoporotic (10% of the sample). So, it's not surprising that the binary BMD variable (osteoporotic yes/no) does not improve predictive accuracy. What happens if you include a 3-level categorical variable for BMD: normal, osteopenic, osteoporotic? This seems like the more natural way to treat BMD as a categorical variable than a simple osteoporotic/not osteoporotic split.	Thank you for this comment and your interest. We did also consider this analysis. However, given the low event rate, further categorisation would have made analysis difficult to interpret. However, we have acknowledged this as an area for future analysis and have added this as potential future research in the discussion ('Discussion – Future Research').	p17
4. Under "Model development," this needs to be rephrased: "Adding binary and continuous BMD to standard risk factors led to a 75% increased hazard of	Thank you for this comment. We apologise for the incorrect phrasing of this sentence. We have now amended this in the main text.	p11

fracture, whilst adding continuous BMD T-score led to a 40% lower hazard per SD improvement...” Though I understand the intent, in fact, “adding variables to the model” does not increase a person’s hazard of fracture.		
5. Table 1: need to give units for alcohol consumption as <3 and >3 units PER DAY rather than just <3 and >3 units.	Thank you for this comment. We have amended this in the table.	Table 1
6. Please reduce the number of decimal places in Tables 1, 3 and 4, e.g., .759 rather than .7592 and 36.7% rather than 36.73%. This makes the tables easier to read.	Thank you for this suggestion. We have reduced the number of decimal places in these tables to help with their readability.	Table 1, Table 3, Table 4
7. What were the standard errors and p-values for the NRIs?	Thank you for this comment. A limitation of the reclassification table analysis is that its metrics are descriptive and methods to test and calculate p-values and standard errors are not yet robust (6-8). Thus SEs and p-values were not calculated.	-
8. The NRI for events can be interpreted as the increase in sensitivity and the	Thank you for this comment and interpretation. We have amended to text to include this explanation (‘Results –	p14

NRI for non-events can be interpreted as the increase in specificity. Please note this for readers to help draw attention to the fact that the inclusion of BMD is helping to improve specificity but not sensitivity.	Reclassification’).	
9. The authors have done a nice job discussing the issue of generalizability in their discussion. I’d like to see a little more discussion about the tradeoffs between slightly improved accuracy and cost. The standard model does pretty well with variables that are inexpensive to measure. This is largely because BMI is tightly linked to BMD, so including BMI in the model gives indirect information about BMD. Is the added	Thank you for this comment. This study is a part of a PhD that developed the methodology to incorporate an economic evaluation to assess the added value of BMD to fracture risk prediction. We are planning a further publication of the economic appraisal of added value of BMD.	-
cost of BMD screening offset by the improved specificity of the new model? Some further discussion of this is warranted.		

Reviewer 4: Statistics in the paper were thorough. Good checking of modeling assumptions in Survival analysis, and good practice of providing estimates, confidence intervals and p-values throughout. However, the interpretation of model fit quality differences seems to be over-played.	Thank you for this comment. We have tried to present the results and facts as they are. As described above a change of 1% was found in Harrell's C-Index which related to 57% of patients being reclassified into their correct risk groups. In line with reviewer's comments, we have rephrased the abstract, discussion (summary of findings), and conclusion to avoid over-playing the interpretation of the results. Further we have also removed the 2 data driven models in the analysis.	p3, p4, p7, p11, Table 2, Table 3, Table 5, p15-p17
1. P 7. line 21/22 looks like there is a typographical error around the word "randomly"	Thank you for this comment. We have amended the text.	p7
2. P 11 line 32/33 -- HR and CI needs an extra decimal point of accuracy (currently from rounding, the HR could be between 1.015, to 1.025; a 60% range!) -- or perhaps even better for clinical relevance would be to provide	Thank you for this comment. We have presented results according to the BMJ Open house style and to keep consistency across the paper. We have also kept in line with the number the decimal places given in other similar studies.	-

HR per 10 year difference in age.		
3. P 11 line 41 -- Do not use insignificant to mean did not reach statistical significance. Not statistically significant just means there is not enough evidence to reject the null hypothesis; but absence of evidence is not the same as evidence of absence. The authors would need to argue that the estimated HR and the complete range of the confidence interval are too small to be clinically relevant in order to argue insignificance.	Thank you for this comment. Although concerning text has been removed in light of other comments, we have ensured this description is used in relevant parts of the paper.	p11
4. P 13 lines 10 and 11 -- based on the results I don't think this statement has been conclusively demonstrated and in particular for BMDcat P 13 lines 13 to 16 -- authors need to note that these differences do not represent reliable improvements of one model over another. The differences between models are very small compared to	Thank you for this comment. We have tried to present the results from the analysis as factually as possible in this section, and agree that changes are small and the confidence intervals for Harrell's C- Index overlap between models. We have commented on this in the discussion ('Discussion – Methodology') and have moderated the implications of these results abstract, discussion (summary of	p3, p15, p17

the width of the confidence intervals.	findings), and conclusion of the paper.	
5. P 13 lines 41/42 I don't think this has been demonstrated.	Thank you for this comment. We have amended this sentence in line with reviewer's comment.	p13
6. P 13 line 40 -- NRI of 0.03 is small -- could it even be a chance improvement? Table 4 doesn't seem to show an overwhelming improvement. Similarly Table 5 NRIs are all small P 15	Thank you for this comment, in line with other comments above, we have moderated the implications of these results abstract, discussion (summary of findings), and conclusion of the paper.	p3, p15, p17

line 7/8 I think this conclusion is overstated.		
7. P 16 line 12/13 -- ratio should be 1:2 rather than 1:3 based on earlier text. (2/3 derivation and 1/3 validation)	Thank you for this comment. The text has been amended to reflect this	p16

References:

1. Soles G, Ferguson T. Fragility fractures of the pelvis. *Curr Rev Musculoskelet Med.* 2012;5:222-8.
2. Abrahamsen B, Stilgren LS, Hermann AP, Tofteng CL, Barenholdt O, Vestergaard P, et al. Discordance between changes in bone mineral density measured at different skeletal sites in perimenopausal women-- implications for assessment of bone loss and response to therapy: The Danish Osteoporosis Prevention Study. *J Bone Miner Res.* 2001;16(7):1212-9.
3. Vickers AJ, Cronin AM. Everything You Always Wanted to Know About Evaluating Prediction Models (But Were Too Afraid to Ask). *Urology.* 2010;76(6):1298-301.
4. Janes H, Pepe MS, Gu W. Assessing the Value of Risk Predictions by Using Risk Stratification Tables. *Annals of Internal Medicine.* 2008;149(10):751-W162.
5. Ridker PM, Paynter NP, Rifai N, Gaziano JM, Cook NR. C-Reactive Protein and Parental History Improve Global Cardiovascular Risk Prediction The Reynolds Risk Score for Men. *Circulation.* 2008;118(22):2243-4.

6. Kerr KF, Wang Z, Janes H, McClelland RL, Psaty BM, Pepe MS. Net reclassification indices for evaluating risk prediction instruments: a critical review. *Epidemiology*. 2014;25(1):114-21.
7. Pencina MJ, D'Agostino RB, Vasan RS. Evaluating the added predictive ability of a new marker: From area under the ROC curve to reclassification and beyond. *Statistics in Medicine*. 2008;27(2):157-72.
8. McGeechan K, Macaskill P, Irwig L, Liew G, Wong TY. Assessing New Biomarkers and Predictive Models for Use in Clinical Practice A Clinician's Guide. *Archives of Internal Medicine*. 2008;168(21):2304-10.

VERSION 2 – REVIEW

REVIEWER	John Kornak University of California, San Francisco, USA
REVIEW RETURNED	29-Nov-2017

GENERAL COMMENTS	Unadjusted model results still focus on p-values. Instead, estimates and confidence intervals should be given along with p-values. Regarding accuracy (original comment 2 re: original p 11, line 32/33. The authors response is not satisfactory here because of the massive uncertainty induced by rounding. Does the journal really insist on only 2 decimal places if a hazard ratio corresponds to only a 0.1% increase? In that case the HR would have to be given as 1.00 when the true value is 1.001. Scientific relevance has to come before following general guidelines. The first sentence under Reclassification is still too strong. The differences are tiny compared to the width of the CIs and so by no means constitute a suggested improvement. A more appropriate statement would be along the lines of e.g. "C-index was estimated to be slightly higher with continuous BMD but this increase is far from conclusive given the wideoth of the confidence intervals"
--

REVIEWER	Eugene McCloskey University of Sheffield, UK
REVIEW RETURNED	30-Nov-2017

GENERAL COMMENTS	The authors have addressed many of the points raised by the reviewers. My only concern is that the authors state that gender specific T-scores are used "in accordance to FRAX". In FRAX, the T-score for MEN and women is derived from the NHANES female reference range. This would lead to a smaller group of men being categorised as osteoporotic - it is likely to have little impact on the overall results and conclusions, but as stated and analysed the data are not in accordance with FRAX.
--

REVIEWER	Kristin Sainani Stanford University, USA
REVIEW RETURNED	05-Dec-2017

GENERAL COMMENTS	The authors have adequately addressed my concerns in their revision.
--

REVIEWER	José António Pereira da Silva Faculty of Medicine, University of Coimbra. Portugal
REVIEW RETURNED	16-Dec-2017

GENERAL COMMENTS	The authors have addressed the reviewers concerns appropriately and in great detail. I just want to bring a recently published paper to their attention, so that they decided whether it should be referenced and discussed: RMD Open. 2017 Sep 26;3(2):e000509. doi: 10.1136/rmdopen-2017-000509. eCollection 2017. Do we need bone mineral density to estimate osteoporotic fracture risk? A 10-year prospective multicentre validation study. Marques A1,2, Lucas R3, Simões E4, Verstappen SMM5,6, Jacobs JWG7, da Silva JAP1. (Attached- Please contact the publisher for full details.)
--

VERSION 2 – AUTHOR RESPONSE

Responses to Reviewers Comments

Comment	Response	Page No
Reviewer 1: José António Pereira da Silva		
The authors have addressed the reviewers concerns appropriately and in great detail	Thank you for you valuable comments.	
I just want to bring a recently published paper to their attention, so that they decided whether it should be referenced and discussed: RMD Open. 2017 Sep 26;3(2):e000509. doi: 10.1136/rmdopen-2017-000509. eCollection 2017. Do we need bone mineral density to	Thank you for bringing this paper to our attention. It is good to see another paper reporting similar findings to our study. We have discussed and referenced this paper in our discussion (Discussion – Summary of Findings, and Discussion – Strengths and Limitations – Answering Evidence Gap).	p16

estimate osteoporotic fracture risk? A 10-year prospective multicentre validation study. Marques A1,2, Lucas R3, Simões E4, Verstappen SMM5,6, Jacobs JWG7, da Silva JAP1.		
Reviewer 2: Eugene McCloskey		
The authors have addressed many of the points raised by the reviewers.	Thank you for you valuable comments.	
My only concern is that the authors state that gender specific T-scores are used "in accordance to FRAX". In FRAX, the T-score for MEN and women is derived from the NHANES female reference range. This would lead to a smaller group of men being categorised as osteoporotic - it is likely to have little impact on the overall results and conclusions, but as stated and analysed the data are not in accordance with FRAX.	Apologies for the confusion with this comment. When stating 'in accordance with FRAX' we meant t-scores were calculated for men and women separately, not that NHANES reference ranges were used. Though we responded to your comment with this statement, we did not add this level of detail to the manuscript. We hope the description in the manuscript better reflects the data and methods of the study.	
Reviewer 3: Kristin Sainani		
The authors have adequately addressed my concerns in their revision	Thank you for you valuable comments.	
Reviewer 4: John Kornak		
Unadjusted model results still focus on p-values. Instead, estimates and confidence intervals should be given	Thank you for this comment. We have added the additional information in the relevant text.	p11

along with p-values		
Regarding accuracy (original comment 2 re: original p 11, line 32/33. The authors response is not	Thank you for re-iterating this point. On hindsight and reflection, we agree that given the small increases in hazard ratios an additional decimal	p12

satisfactory here because of the massive uncertainty induced by rounding. Does the journal really insist on only 2 decimal places if a hazard ratio corresponds to only a 0.1% increase? In that case the HR would have to be given as 1.00 when the true value is 1.001. Scientific relevance has to come before following general guidelines.	place would be more informative. Rather than add this to the text however, we have added this to Table 2 for consistent presentation of all estimates. We hope this addresses your comment.	
The first sentence under Reclassification is still too strong. The differences are tiny compared to the width of the CIs and so by no means constitute a suggested improvement. A more appropriate statement would be along the lines of e.g. "C-index was estimated to be slightly higher with continuous BMD but this increase is far from conclusive given the wideoth of the confidence intervals"	Thank you for your comment and suggestion, we have further moderated the tone of this sentence. We appreciate your example sentence; it has been discussed in detail by the co-authors. As written, it appears to be an interpretation of the results and so we have included it in 'Summary of Findings' section of the Discussion.	p14, p16

VERSION 3 – REVIEW

REVIEWER	John Kornak University of California, San Francisco. USA
REVIEW RETURNED	14-Feb-2018

GENERAL COMMENTS	The authors added HR and CI's for unadjusted results which is great. However, there are still two outstanding issues and the first one below is critical. 1) The text immediately under the Reclassification heading is still misleading as I said before. Putting qualifying text elsewhere in the paper does not mitigate this and neither does changing the word "showed" to "indicated". I do not believe it indicates this given the tiny magnitude difference coupled with massive uncertainty. 2) The needed accuracy for the hazard ratios needs to be updated in the text, not just in tables.
--

REVIEWER	Eugene McCloskey University of Sheffield, UK
REVIEW RETURNED	24-Feb-2018

GENERAL COMMENTS	I am happy with the final version of the paper and no further comments to add.
--

VERSION 3 – AUTHOR RESPONSE

Dear Sir/Madam,

Thank you for your conditional acceptance of our original article titled, "Does Bone Mineral Density Improve the Predictive Accuracy of Fracture Risk Assessment? A Prospective Cohort Study in Northern Denmark".

We are pleased to have satisfied most of the reviewers and have further amended the manuscript in accordance to comments raised by Reviewer 4. Specifically, we have added more precision to the estimates in the text, and have again moderated the tone of the paper. With respect to the latter, this is in addition to the revisions from the previous resubmission, where we have also amended the discussion to emphasise the uncertainty of the results.

A comment not appropriate to be relayed to Reviewer 4:

We hope we have satisfied Reviewer 4 at this point. Although we anticipate further contention with this reviewer, we are pleased that other referees are happy with the wording of text under question and hope no further amendment is required.

We hope that the changes we have made to fulfil your requests, but please do let us know if you would like us to make any further changes.